# Graphene Nanoplatelets-Based Textured Polymeric Fibrous Fabrics for the Next-Generation Devices

**DOI:** 10.3390/polym14245415

**Published:** 2022-12-10

**Authors:** Enrica Chiesa, Erika Maria Tottoli, Alessia Giglio, Bice Conti, Mariella Rosalia, Laura Giorgia Rizzi, Rossella Dorati, Ida Genta

**Affiliations:** 1Department of Drug Sciences, University of Pavia, Viale Taramelli 12, 27100 Pavia, Italy; 2Directa Plus S.p.a., COMO NexT, Via Cavour, 2, 22074 Lomazzo, Italy

**Keywords:** graphene nanoplatelets, thermoplastic polymer, polylactic acid, electrospinning, texturized fabric

## Abstract

Graphene is a 2D crystal composed of carbon atoms in a hexagonal arrangement. From their isolation, graphene nanoplatelets (nCD) have revolutionized material science due to their unique properties, and, nowadays, there are countless applications, including drug delivery, biosensors, energy storage, and tissue engineering. Within this work, nCD were combined with PLA, a widely used and clinically relevant thermoplastic polymer, to produce advanced composite texturized electrospun fabric for the next-generation devices. The electrospinning manufacturing process was set-up by virtue of a proper characterization of the composite raw material and its solution. From the morphological point of view, the nCD addition permitted the reduction of the fiber diameter while the texture allowed more aligned fibers. After that, mechanical features of fabrics were tested at RT and upon heating (40 °C, 69 °C), showing the reinforcement action of nCD mainly in the texturized mats at 40 °C. Finally, mats’ degradation in simulated physiological fluid was minimal up to 30 d, even if composite mats revealed excellent fluid-handling capability. Moreover, no toxic impurities and degradation products were pointed out during the incubation. This work gains insight on the effects of the combination of composite carbon-based material and texturized fibers to reach highly performing fabrics.

## 1. Introduction

Graphene is a two-dimensional (2D) sheet of covalently bonded carbon atoms with sp^2^ hybridization, thickly packed into a hexagonal structure; it is the first 2D material ever made by mankind, with a thickness of one atom (0.34 nm) [1]. Thanks to its structure, it has unique features such as (i) thinness; (ii) electric conductivity; (iii) thermal conductivity, being 10-fold higher than diamond, the best thermal conductor so far [2]; (iv) optimal mechanical properties with better strength and elasticity than steel, as shown by Lee and coworkers [3]; and (v) gas impermeability, including Helium [4]. Graphene forms the basis of the “graphene family nanomaterials”, a comprehensive category of materials using graphene as the precursor; among these, the most famous are pristine graphene, few-layer graphene, graphene oxide, reduced graphene oxide (rGO), and graphene nanosheets [5]. This category includes nanomaterials which differ for the number of layers, chemical composition, defects density, and surface area [5]; such nanomaterials are appealing thanks to the planar structure, light weight, high aspect ratio, electrical conductivity, and mechanical toughness. Therefore, the intrinsic properties of graphene and its derivative, expected to exceed that of any other material, stimulate the use of carbon-filler in advanced composites to achieve next-generation devices, potentially useful for photonics, opto-electronics, protection coatings, gas barrier films, and biomedical fields [6,7].

Herein, pristine graphene nanoplatelets (nCD) produced by a patented continuous process [8] were used. It was the purest graphene mass-produced so far, and it consists of a mixture of single layer and few layers of graphene with a thickness range from 0.34 to 100 nm [9]. nCD became an attractive alternative to common nanostructured fillers such as metallic nanoparticles and clays, since nCD can be easily and successfully blended with polymers which are probably the most important class of material, having driven the industrial revolution [10,11,12].

In this work, nCD were combined with polylactic acid (PLA), which is recognized as the second most-traded polymer worldwide, and its market is going up by more than 10% annually [12].

Furthermore, the sharp rise in the use of PLA in the biomedical field is clearly showed by the increasing number of related publications: an average of around 1400 papers per year over the past four years (Pubmed; search term: polylactic acid).

Herein, the advantages derived by the raw materials (nCD and PLA) were brought together with an innovative manufacturing method, electrospinning, to produced micro/nanofibrous mats. The electrospinning technique was defined by the European Commission as a key enabling technology, and the advanced manufacturing process allowed the superior fabric’s properties. Electrospun matrices revealed a high specific surface as well as tunable mechanical properties and topological features due to the fibrous structure, and they successfully found application in drug delivery [13,14,15], tissue engineering [16,17], oil–water separation [18], and air filtration [19].

The combination of nanocarbons into electrospun polymeric mats was nicely reviewed by Scaffaro et al. (2017), where it was noted that most of the experimental works involved carbon nanotubes and graphene oxide [20]. 

Few studies on electrospun matrices, mainly composed of randomly oriented fibers blending PLA and nCD, were reported in the literature, showing controversial results [21,22,23,24]. 

Scaffaro et al. (2017) fabricated electrospun membranes and thin films integrating nCD into a PLA solution. They showed that composite films prepared by solvent casting improved strengthening, stiffening, and toughening effects, while a good ductility was preserved. Instead, in the fibrous membranes, nCD damaged fibers’ morphology and their stiffness was compromised, whereas the tensile strength increased as long as the nCD content increased [22]. Moreover, ternary composite nanofibers were fabricated by Yang et al. (2016) by using electrospinning technology to blend nCD and multi-walled carbon nanotubes into the PLA matrix. They obtained randomly oriented uniform fibers where the crystal structure of PLA was upheld. Finally, the addition of nanocarbon fillers improved the thermal stability, and the fabrics’ degradation slowed down [24]. Recently, Leones et al. (2021) investigated the reinforcement effect of different organic and inorganic nanofillers into PLA nanofibers. The addition of nCD led to randomly oriented fibers in the micrometer range where the crystallinity of the PLA was significantly reduced if compared to the neat polymer. In this case, mechanical tests on composite fabrics revealed reduced elastic modulus and comparable tensile strength and elongation at break, with respect to the neat PLA mat [21]. 

With this work, innovative texturized fabrics were produced through a patented process [25] to create flexible, highly absorbent, and conformable fabrics. The aim was to systematically optimize the manufacturing, starting from the characterization of the raw composite material and its solutions. Moreover, the effects of the advanced combination of nCD and fabrics’ topology were studied to address the production of the next-generation devices. To the best of our knowledge, this is the first attempt to assess the relationship between the topology of electrospun fabrics blending PLA/nCD and the matrix’s properties. 

This work was intentionally a wide characterization of the new mats considering all the potential advantages derived from nCD; however, the biomedical application was preliminary and explored by testing the integrity of the matrix in simulated physiological fluid.

## 2. Materials and Methods

### 2.1. Materials

PLA blended with pristine graphene nanoplatelets (PLA-nCD, 1PLA-GRAF2 wire-id 1.75 mm, GRAFYLON^®^ 3D filaments) and PLA (1PLA00005 transparent wire-id 1.75 mm) were gifted by Directa Plus S.p.A. (Lomazzo, Italy). GRAFYLON^TM^ resulted from a collaboration between Filoalfa and Directa Plus companies and the filament contains Graphene Plus (1% wt), a nanomaterial made up of pristine graphene nanoplatelets (nCD). Pristine graphene nanoplatelets (nCD) (produced by Directa Plus) are characterized by a thickness of 2–4 nm and a lateral dimension of 10–15 µm [8].

Methylene chloride (MC), N,N-dimethylformamide (DMF), and tetrahydrofuran (THF) analytical grade were supplied by Carlo Erba (Milan, Italy). Dulbecco’s Modified Eagle’s Medium—high glucose (DMEM), Dulbecco’s phosphate buffered saline (PBS 10×, sterile), penicillin–streptomycin solution (100×), dimethyl sulfoxide (DMSO), and 3-(4,5-dimethylthiazol-2-yl)-2,5-diphenyltetrazolium bromide were obtained from Sigma Aldrich (St. Louis, MO, USA). Normal human dermal fibroblasts (NHDFs) were provided by PromoCell (Heidelberg, Germany); the cells were cultured at 37 °C with 5% CO_2_ in **a** 25 cm^2^ culture flask with DMEM containing 1% (*v*/*v*) penicillin–streptomycin and 10% (*v*/*v*) FBS.

### 2.2. Polymer Characterization

PLA-nCD and PLA were characterized by gel permeation chromatography (GPC) and differential scanning calorimetry (DSC).

GPC (1260 Infinity GPC apparatus, Agilent technology, Santa Clara, CA, USA) was used to assess the polymer molecular weight. The chromatographic system consisted of three PhenoGEL columns (5-µm particle size—500-Å, 103-Å, 104-Å pore size) and a refractive index detector. THF was the mobile phase and a flow rate of 1 mL/min was used. PLA-nCD and PLA wires were dissolved in THF at 1 mg/mL, nCDs were removed by centrifugation, and the polymer solutions were filtered throughout 0.45 µm nylon filters (Whatman Uniflo, GE Healthcare, Pittsburgh, USA) and analyzed. The PLA molecular number, molecular weight, and polydispersity index were assessed against polystyrene standards with molecular weight ranging from 1480 to 361,500 Da. 

The glass transition, crystallization, and melting behaviors of PLA and PLA-nCD were evaluated by differential scanning calorimetry (DSC, DSC Q2000 apparatus interfaced with a TA 5000 data station, TA Instrument, New Castle, DE, USA). Samples (11–15 mg), hermetically sealed in aluminum pans, underwent cyclic measurements under nitrogen flow (50 mL/min). Analyses were composed by a first heating to 180 °C at the heating rate of 5 K/min, a cooling down to 20 °C at 5 K/min, and a second heating up to 180 °C at 2 K/min. The temperature and enthalpy change of the cold crystallization (Tcc, ∆Hcc) and of the melting (Tm, ∆Hm) were taken from the DSC curve of the second heating.

### 2.3. Polymer Solution Characterization

PLA and PLA-nCD were dissolved by pure MC and a mixture of MC:DMF of 90:10 (v:v). Different polymer concentrations ranging from 4% to 24% *w/v* were tested. The rheological behavior of the polymer solution was investigated by a Kinexus Plus rotational rheometer (Alfatest, Milan, Italy), and equipped with a temperature control system and a plate-cone geometry (CP4/40, cone angle 1°, diameter 40 mm, gap of 0.15 mm). During the analysis, a solvent trap was used to avoid MC evaporation.

A shear rate ramp test was carried out to measure the viscosity of the polymer solution at an increasing shear rate (0.1–1000 s^−1^).

All the rheological measurements were performed at 25 °C.

### 2.4. Textured Fabrics Preparation

Textured fabrics were prepared using electrospinning Nanon-01A (MEEC Instruments, ltd., Ogori-shi, Fukuoka, Japan) equipped with a dehumidifier. PLA-nCD and PLA solutions were solubilized in both MC and MC:DMF (90:10 v:v) at 20% *w*/*v* and then sonicated to removed air bubbles. The polymer solution was fed through a 20G needle and the distance between the needle and the collector was maintained constant at 15 cm. Flow rates of 0.1 and 0.3 mL/h were tested while the voltage varied from 20 to 25 kV. Spinneret oscillation (30 mm) and speed (1 mm/sec) were kept constant. The electrospinning time was 20 min. Different texturized fiber dressings were prepared using aluminum wire conductive plates placed on a collector and having specific textures: horizontal diamond mesh (sample code: D-PLA or D-PLA-nCD), circle rhombic mesh (sample code: R-PLA or R-PLA-nCD), and horizontal square mesh (sample code: F-PLA or F-PLA-nCD), as disclosed in PCT WO2021064673 [25].

Electrospun samples were stored in a dryer at 10 °C, 43% RH.

### 2.5. Morphological Characterization of the Textured Fabrics

A morphological characterization by scanning electron microscopy (SEM) allows the investigation of the microstructural features of the fibrous fabrics. Samples were observed by Zeiss EVO MA10 (Carl Zeiss, Oberkochen, Germany), and each sample was coated with gold in an argon atmosphere. Images were obtained at high voltage (20 kV), in high vacuum, at room temperature, and at various magnifications 1k×. All SEM images were analyzed in terms of fiber diameter, fiber orientation, and porosity percentage by ImageJ software (*n* = 50). 

### 2.6. Mechanical Properties 

The tensile mechanical properties of fabrics were examined by a mechanical tester machine Mark-10 ESM303 (Force Gauge Model MI5-5, G1013; Copiague, NY, USA) equipped with a 25-N load cell and MESUR gauge Plus software. A dog bone-shaped prototype (80 × 10 × 4 mm) was used according to the ASTM (American Society for Testing Materials). A tensile test were carried out at a constant speed of 10 mm/s [26]. Tensile properties were tested on PLA and PLA-nCD fabrics stored at RT, 40 °C (below the Tg of PLA) and 69 °C (above the Tg of PLA) for 3 h and 6 h. The Young’s modulus and elongation-at-break (breaking point) were determined, and results were expressed as average ± standard deviation. 

### 2.7. In Vitro Degradation Study

The electrospun matrices’ degradation behavior was evaluated in a simulated physiologic condition, and it was performed as follows: round-shaped composite mat samples (1.8 cm^2^) were incubated in 5 mL of DMEM with FBS (10% *v*/*v*) and antibiotics (1% *v*/*v*) at 37 °C up to 30 d.

At each scheduled time point the incubation medium was withdrawn, and the electrospun matrices were collected and characterized for their fluid adsorption and mass loss (Section 2.7.1); PLA molecular weight was determined as Section 2.2. Incubation media were tested on NHDFs to evaluate the potential toxicity of the fabric’s impurities and degradation products (Section 2.7.2).

#### 2.7.1. Fluid Adsorption and Mass Loss

PLA and PLA-nCDs fabrics were weighted (M0) and then incubated at 37 °C with 5 mL of DMEM containing FBS and antibiotics. On the 7th and 30th day of incubation, the wet mats were washed three times and weighted wet (Mt). The samples were freeze-dried overnight (−48 °C, 0.2 bar) (Lio5P, 5Pascal, Milano, Italy), and the freeze-dried samples were weighted (Mx). Fluid uptake percentage and mass loss percentage were calculated as stated in Equations (1) and (2), respectively:Fluid uptake (%) = 100 × (Mt − M0)/M0 (1)
Mass loss (%) = 100 × (M0 − Mx)/M0(2)

Three specimens were tested, and the results were reported as mean ± standard deviation (SD).

#### 2.7.2. Cytotoxicity of the Fabrics’ Degradation Products

A cytotoxicity test was performed on the collected incubation media as in Section 2.7.

The test was carried out by using NHDF as a model line by following the protocol described in Chiesa et al. 2020 (Chiesa 2020). NHDFs were seeded at the density of 1 × 10^4^ and after 24 h they were treated with 0.2 mL of incubation media for 24 h. Cell viability was assessed by MTT assay, and NHDFs incubated with fresh media were used as the control (CTRL). Results were displayed as the cell viability percentage compared to CTRL.

### 2.8. Statistical Analysis

A statistical analysis tool was carried out by GraphPad Prism 6 software (GraphPad software, La Jolla, CA, USA). A one-way analysis of variance (ANOVA) with either Sidak’s or Tukey’s multiple comparison analysis was performed.

## 3. Results

### 3.1. Polymer Characterization

PLA and PLA-nCD wires were characterized in terms of molecular weight by GPC. PLA wire showed an average molecular weight of 193,304 Da, an average molecular number of 103,764 Da, and a PI value of 1.86. No significant differences were revealed by using PLA-nCD. The PLA-nCD average molecular weight and average molecular number were 211,865 Da and 146,398, respectively (PI value = 1.45). The glass transition, crystallization, and melting behaviors of PLA and PLA-nCD were pointed out by DSC curves recorded during the second heating (Appendix A), and the thermal parameters are shown in Table 1. The Tg values of PLA and PLA-nCD did not change significantly (56.7 °C and 56.9 °C, respectively), indicating that the reinforcement of nCD did not influence the plastic polymer behavior. Moreover, PLA exhibited a bimodal endothermic peak at 147 °C and 155 °C, indicating the formation of an unstable crystal during the cold crystallization. The same characteristic was observed in the DSC curve of PLA-nCD (T_m1_ = 145 °C; T_m2_ = 153 °C) and the presence of nCD did not change the T_m_ as previously demonstrated by using a physic blend of PLA_PCL and nCD. A cold crystallization exothermic peak was observed at 105 °C for the neat PLA wire, and at 98 °C for the composite material. As can be noted, the T_cc_ decrease in presence of nCD suggests that the inorganic filler altered the PLA nucleation. 

### 3.2. Polymer Solution Characterization

The aim was to identify the entanglement concentration that is the boundary conditions between the semidilute unentangled and semidilute entangled concentration regimes, and it is defined as the point at which the polymer chain overlapping occurs and it constrains the chain motion. A shear rate ramp from 0.1 to 1000 s^−1^ was performed on PLA and PLA-nCD solutions in a concentration rank of 4–24% *w*/*v*. The analysis was carried out at 25 °C. Both polymer wires were solubilized in MC and MC:DMF blend. Figure 1 plots the shear rate viscosity as a function of the shear rate value, and as can be noted, all the polymer solutions (obtained by MC as a solvent) showed Newtonian behavior over the whole range of the shear rate, and only a slight reduction of the viscosity along the shear rate occurred. As shown in Figure 1, the shear rate viscosity progressively increased if the polymer concentration arose. At the 10 s^−1^shear rate, the solution viscosity increased from 0.02 Pa × s to 12.3 Pa × s for the PLA-nCD solution at 4% and 24%, respectively, and from 0.02 Pa × s to 3.05 Pa × s for PLA solutions at the same concentration. 

The zero shear rate viscosity was determined experimentally and by extrapolating the value from a shear rate sweep experiment [27], and the results were reported in Table 2. By increasing the polymer concentration from 4 to 24% *w*/*v*, the zero shear rate viscosity increased from 0.022 to 7.87 Pa × s and from 0.025 to 10.3 Pa × s for PLA and PLA-nCD, respectively.

The polymer contribution to the zero shear rate viscosity was studied by defining the specific viscosity by using the following equation (Equation (3)).
η_sp_= (η_0_ − η_s_)/ η_s_(3)
where η_0_ is the zero shear rate viscosity of the PLA-nCD solution at a specific concentration, and η_s_ is the DCM viscosity tabulated and reported in the literature (η_s_ = 0. 44 mPa × s) [28].

The calculated η_sp_ were plotted as a function of the concentration (Figure 2).

Changes in the slope highlighted the onset of the several concentrations’ regimes. The semidilute unentangled regime was revealed from 4% to 8% *w*/*v*, while the semidilute entangled regimes were in a rank of 8–20% *w*/*v*. Finally, the concentrated regime was found for a concentration higher than 20% *w*/*v*. In this study, the dilute region was not identified since the rheological measurement was not carried out for a polymer solution lower than 4% wt. Since the fiber formation by electrospinning was theoretically ensured for the polymer solution belonging to the entangled region, a concentration of 20% *w/v* was selected for further study. The addition of DMF (10% *v*/*v*) did not reveal evident changes in the viscosity values; Figure 3 shows superimposable viscosity patterns for both solvent MC and MC:DMF. 

### 3.3. Textured Fabric Preparation

At first instance, PLA-nCD flat fabrics (PLA-nCD-F) were electrospun starting from 20% *w/v* polymer solution, and the influence of the process parameters as voltage and flow rate on the fabric’s morphology was evaluated.

The fibrous matrices were observed by SEM (Figure 4) and were elaborated by ImageJ software (1.52a, NHI); results are reported in Figure 5. 

As shown in Figure 5A, an increase in the feed rate from 0.1 to 0.3 mL/h triggered an increase in the fiber’s diameter from 2.32 ± 0.4 μm to 3.38 ± 0.3 μm (*p* value < 0.05), with 20 kV voltage and from 2.17 ± 0.10 μm to 3.87 ± 0.62 μm (*p* value < 0.01) with 25 kV voltage. The total porosity percentage ranged from 45 ± 3% to 61 ± 3% for all the samples; however, a slight increase (*p* value< 0.05) of the porosity percentage was observed by increasing the flow rate at 0.3 mL/h. The flow rate also affected the mean pore area (Figure 5c respectively). At the voltage of 20 kV, the mean pore area significantly increased from 218 ± 9.33 µm^2^ to 555 ± 60.1 µm^2^ (*p* value < 0.001) by using the 0.1 and 0.3 mL/h flow rate, respectively. The same behavior was observed at 25 kV, where the mean pore area was 135 ± 32.6 µm^2^ and 499 ± 122 µm^2^ for matrices obtained at 0.1 and 0.3 mL/h. 

No significant impact of the voltage on the fiber diameter was noted. 

Further textured fabrics were produced by setting the electrospinning apparatus at 20 kV voltage and 0.3 mL/h flow rate. Figure 6 displays the macro details of the PLA-nCD and PLA texturized fabrics, while SEM images of the textured fabrics and images’ elaboration to assess fiber size and orientation and fabric porosity were reported in Appendix A. All prototypes showed smooth, beadless, and interconnected fibers.

Table 3 provides an overview of texturized fabrics’ characteristics related to mean diameter, fiber orientation, and porosity.

By using MC as a solvent, F-PLA had fibers’ mean diameter of 3.35 ± 0.44 µm, and the size distribution of the fibers was characterized by a mode of 2.83 µm and a median of 3.48 µm. In total, 30% of the fibers had an equal orientation between 70 and 45 degrees. Regarding the porosity, F-PLA matrices presented a porosity percentage of 45% and a mean pore area of 89 µm^2^. R-PLA showed a microfiber of 4.64 ± 1.22 µm, and the size distribution was characterized by a mode of 3.49 µm and a median of 5.23 µm. The percentage of fibers with equal orientation was 33%. Moreover, the R-PLA porosity was 30% with a mean pore area of 73.6 µm^2^.

S-PLA had fibers of 3.85 ± 0.32 µm; median and mode were 3.27 µm and 3.92 µm; 36% of the fibers was oriented between 30 and 49 degrees. S-PLA had a percentage porosity of 31% with a mean pore equal to 50 µm^2^.

D-PLA was composed of microfibers of 4.89 ± 0.84 µm (mode = 4.58 and median = 5.23 µm). In total, 27% of the fibers of the protype had a similar orientation, and the porosity was decreased since D-PLA porosity percentage was 21% with a mean pore area of 44 µm^2^. 

The addition of nCD caused an overall decrease in the mean diameter of the fibers (Table 3). Fibers of F-PLA-nCD showed an average diameter of 2.85 ± 0.10 µm with a mode and median of 2.61 µm and 3.05 µm, respectively. Furthermore, 28% of the fibers was oriented between 30 and 49 degrees. Flat matrix presented a porosity percentage of 50% and the mean pore area was 90.6 µm^2^.

R-PLA-nCD had fibers of 3.21 ± 0.41 µm; fibers’ size distribution was characterized by a mode of 2.61 µm and a median of 3.49 µm. Furthermore, 23% of the fibers were oriented between 30 and 49 degrees. The porosity percentage was 52% showing a mean pore area of 123 µm^2^. S-PLA-nCD showed fibers with a mean diameter of 2.88 ± 0.04 µm, a mode of 2.39 µm, and a median 3.27 µm. Prototypes showed a porosity of 48.2% and a mean pore area of 79.5 µm^2^.

D-PLA-nCD were formed by fibers of 3.27 ± 0.05 µm (mode = 2.61 and median = 3.27 µm). In addition, 53% of the fibers had the same orientation into the fabric which was characterized by a porosity percentage of 58% and a mean pore area of 178 µm^2^.

Adding DMF (10% *v*/*v*) to the solvent system slightly influenced the fabrics’ characteristics. F-PLA-2 presented very homogeneous fibers having a mean diameter of 2.73 ± 0.02 µm with a size distribution described by a mode of 2.61 µm and a median of 2.62 µm. A small fraction of fibers (26%) had similar orientations in a rank of 50-70 degrees. Interconnected fabrics showed a porosity percentage of 49.9% and a mean pore area of 72.1 µm^2^.

R-PLA-2 fabrics showed fibers’ mean diameter of 3.13 ± 0.62 µm with a mode and median of 2.61 µm and 3.49 µm, respectively. Furthermore, 31% of the fibers showed similar orientation, forming a network with 35.2% of the porosity percentage and a mean pore area of 55.6 µm^2^.

A morphological examination of S-PLA-2 showed aligned fibers with a size of 2.74 ± 0.16 µm with homogeneous size distribution (mode = 2.61 µm and median = 2.83 µm). Most of the fibers (71%) had the same orientation. Regarding the porosity, S-PLA_2 revealed a porosity percentage of 50.4% and a mean pore area of 80.9 µm^2^.

D-PLA-2 was characterized by a fiber’s mean diameter of 2.60 ± 0.02 µm and 48% of the fibers were oriented between 50 and 69 degrees, and these fibers formed pores with a mean pore area of 82.9 µm^2^ (porosity percentage = 48.2%).

As before, the addition of nCD triggered a reduction in the fiber’s mean diameters and a more oriented fibers’ pattern.

More in detail, F-PLA-nCD-2 revealed homogeneous fibers with a size of 2.60 ± 0.16 µm. A fiber’s fraction of 37% had the same orientation from 10 to 29 degrees. The porosity evaluation revealed a percentage porosity of 43.2% and a mean pore area of 49.7 µm^2^.

R-PLA-nCD-2 had mean fiber diameters of 2.66 ± 0.03, with 52.0% of the fibers aligned around 10 degrees. Moreover, the porosity percentage of the fabrics was 46.7% with a mean pore area of 62.8 µm^2^.

The S-PLA-nCD-2 morphological examination showed fibers of 2.60 ± 0.25 µm with a high fiber alignment (63% of the fibers between 30 and 49 degrees). As for the porosity percentage, it was 46.3% with a mean pore area of 92.3 µm^2^.

D-PLA-nCD-2 was characterized by fibers of 2.68 ± 0.16 µm, and mode and median were 2.61 and 2.83 µm, respectively. An important fraction of fibers (60.8%) had the same orientation (31–50 degrees). In terms of porosity, D-PLA-nCD-2 showed a porosity percentage of 41.2% and a mean pore area of 55.6 µm^2^.

### 3.4. Mechanical Properties 

Tensile properties were tested on PLA and PLA-nCD fabrics stored at RT, 40 °C (below the Tg of PLA), and 69 °C (above the Tg of PLA). The elastic modulus and breaking point detected at RT were reported in Figure 7, where it can be noted that the presence of nCD influenced the fabrics’ mechanical properties. The elastic modulus was significantly increased for F-PLA-nCD (flat, *p* value < 0.0001) and D-PLA-nCD (diamond, *p* value < 0.01), with respect to the PLA counterpart. Moreover, flat fabrics showed the highest elastic performance (52.7 kPa) if compared to R-PLA-nCD (21.8 kPa) and S-PLA-nCD (24.6 kPa). As for the breaking point, nCD improved the ability of the fabrics to withstand a tensile force, breaking point of F-PLA-nCD (278 kPa), and R-PLA-nCD (281 kPa) was significantly higher than the pristine counterparts (129 kPa and 114 kPa, F-PLA and R-PLA, respectively). Moreover, D-PLA-nCD (162 kPa) showed an enhanced breaking point value with respect to D-PLA (55 kPa), even if this difference was not statistically significant.

The mechanical properties of the fabrics were tested after incubation at 40 °C for 3 h and results were reported in Figure 8. PLA-nCD showed an increase in the elastic modulus if compared with the PLA counterparts; this boost was observed for all the textures tested and it was statistically significant (*p* value < 0.05) for round, square, and diamond patterns (Figure 8A). Increasing the temperature up to 40 °C, textured fabrics exhibited superior properties in terms of elasticity with respect to the flat matrix. R- PLA-nCD, S-PLA-nCD, and D-PLA-nCD showed an elastic modulus of 87.2 kPa, 88.1 kPa, and 221 kPa, respectively. Regarding the breaking point, there were no significant differences between PLA and PLA-nCD fabrics after heating as observed in Figure 8B. However, the patterns of the fabrics influenced the resistance under stress, and breaking point values of round, square, and diamond fabrics were higher than those of flat fabrics.

Prolonging the incubation time up to 6 h, a drop of the deformation behavior was recorded (Figure 9A); elastic moduli ranged from 2.9 kPa to 9.7 kPa and were detected for all the fabrics tested. All the patterns tested showed similar behavior upon heating for 6 h. Unlike the elastic properties of the samples, no statistical changes of the ability of the prototypes to resist against applied stress were observed. As shown in Figure 9B, by prolonging the incubation time at 40 °C from 3 h to 6 h of incubation, comparable values of the breaking point were recovered. Moreover, S-PLA-nCD and D- PLA-nCD, heated for 6 h at 40 °C, revealed unmodified breaking point values (ns, *p* value > 0.05) compared to those at RT.

After heating at 69° C, PLA-nCD samples were characterized by a marked variability. As shown in Figure 10A, a notable decrease in the elastic modulus values was observed for all the patterned fabrics, and instead no differences were revealed by varying the incubation time (3 h and 6 h, light blue and blue bars, respectively). Regarding the resistance to break (Figure 10B), as observed before, F-PLA-nCD and R-PLA-nCD showed reduced breaking point values compared to those revealed at RT, while the temperature increase did not affect the breaking points of S-PLA-nCD and D-PLA-nCD. 

### 3.5. In Vitro Degradation Study

A degradation study was carried out by incubating the textured fabrics (both PLA and PLA-nCD) in DMEM supplied with FBS and antibiotics for 7 and 30 d. At the end-time points, the mats were weighted to assess the absorbency skill, and then they were freeze-dried to evaluate both the mass loss percentage and polymer molecular weight variation. The results of water adsorption are reported in Figure 11A. Electrospun micro/nanofibers are known to possess a great fluid handling capacity, and all the samples tested showed a water uptake percentage greater than 150%, indicating that after incubation in fluid, the mass of the specimens was at least duplicated at 7d of incubation. More in detail, the water adsorption capability of PLA-nCD mats (WU in a rank of 167–314%) was reduced if compared to neat PLA mats (WU in a rank of 427–603%). This behavior can be ascribed to an increase in hydrophobicity derived by the nCD content. Moreover, by prolonging the incubation up to 30d, an increase in the water uptake was observed for all the specimens, revealing values ranging from 397% to 780%, since the weight of the wet fabrics was at least four times greater than the initial weight of the dry mats. Despite the fact that this enhancement was more evident in PLA-nCD (WU in a rank of 397–637%), the water absorption capability of microfibers with nCD was lower with respect to their PLA counterparts (WU among 577–780%). 

The specimens were then freeze-dried to assess the mass loss percentage (Figure 11B). The mass variation over the time was minimal for all the samples tested; at the 7th day of incubation, the weight of the samples was almost the same as the initial one, and the highest mass loss value was 6.2% (S-PLA-nCDs at 7d). Extending the incubation up to 30d, the mass loss values remained lower than 15% (Figure 9B). The molecular weight of the fabrics that had undergone the degradation process was assessed by GPC. After the electrospinning process, the polymer Mw and Mn were 204,462 ± 6499 Da and 135,877 ± 11,915 Da (PI = 1.51 ± 0.098) for PLA-nCD, and 193,304 ± 5680 Da and 105,233 ± 4534 Da (PI = 1.85 ± 0.025) for PLA. 

No significant differences were highlighted in the first 7d of incubation: PLA-nCD mats showed an Mw of 201,026 ± 9097 Da and an Mn of 138,298 ± 5465 Da, while PLA mats had an Mw of 198,178 ± 10,467 Da and an Mn of 103,237 ± 2955 Da. At the 30th day of incubation, Mw reductions of 19 ± 4% and 13 ± 4% for F-PLA-nCD and F-PLA were detected, respectively (Mw of 162,466 ± 17,526 Da and 172,784 ± 3200 Da). Fiber patterns did not affect the Mw variation during the incubation, and superimposable degradation profiles were observed for PLA-nCD and PLA matrices. As can be noted from Figure 11C, at 7d of incubation, the Mw reduction was 4.3 ± 1.3%, 3.5 ± 2.3%, 4.7 ± 2.9%, and 4.9 ± 1.9% for F-PLA-nCD, S-PLA-nCD, R-PLA-nCD, and D-PLA-nCD, respectively. Furthermore, the Mw of the fabrics slightly decreased at 30d, reaching a variation of 19 ± 4%, 14 ± 3%, 15 ± 2%, and 13 ± 2% for F-PLA-nCD, S-PLA-nCD, R-PLA-nCD, and D-PLA-nCD, respectively. Similar results were obtained with PLA fabrics, regardless of the selected texture.

Media recovered after the incubation were incubated on NHDFs for 24 h to test their toxicity through an MTT assay. As can be noted from Figure 11D, the cell viability was always higher than 70%, indicating the absence of toxic degradation products.

## 4. Discussion

PLA is a thermoplastic aliphatic polyester derived from renewable resources characterized by environmentally friendly properties, excellent biocompatibility, and biodegradability. It is widely used in several fields including regenerative medicine, drug delivery, and sensoring; nevertheless, PLA presents some shortcomings, such as poor mechanical behavior and low bioactivity. Several works attempted to overcome these problems through functionalization and modification. Research on PLA is still ongoing, looking for the optimization and approach of new applications. In this context, fillers can be used to achieve particular and specific functionalities such as antimicrobial properties, thermal conductivity, and mechanical reinforcement. In parallel, the application of graphene grows continuously, and is part of pursuing a new, unexplored field. Many works seek to develop nanocomposites combining graphene or its derivative with numerous polymeric matrices [26,29,30,31,32,33,34,35]. It has been demonstrated that the addition of nCD (4% wt) to PLA-PCL triggered an improvement of the thermal conductivity (1.27 ± 0.008 W/m K) and thermal diffusivity (1.07 ± 0.068 mm^2^/s) compared to the PLA-PCL matrix counterpart (about 0.3 W/m K) [26]. Moreover, the conductivity of a polymer can be greatly enhanced by using graphene as the conductive filler since polymers commonly possess conductivity of about 10^−10^ S/m, unlike the conductivity of graphene, which can reach 10^4^–10^5^ S/m [36]. When the filler concentration reached a threshold, the nanocomposite changed from insulators to conductors, and the electrical conductivity rose by a range of 10 orders of magnitude [37].

Here, the fabrication of PLA-graphene (PLA-nCD) electrospun fibers was investigated and optimized. PLA and PLA-nCD were characterized for their average molecular weight and average molecular number, crystallization, and melting behaviors. PLA and PLA-nCD raw materials revealed comparable average molecular weights. During the second heating process, typical thermal events were observed for both PLA and PLA-nCD as glass transition, cold crystallization, and double melting behavior. A double endothermic melting peak indicates that, during the cold crystallization process, the formation of two different PLA crystals happened; this behavior is commonly observed, and it was explained by Yasuniwa et al. (2004) by the slow rate of crystallization leading to a α′-/α-crystal polymorphism [38]. Since PLA is a typical semicrystalline polymer, a crystallization peak at approximately 98–104 °C was observed. The addition of nCD decreased the Tcc of the composite material compared to the neat PLA. This shift to a lower temperature (around 2 °C) suggests that the cold crystallization behavior of PLA has been enhanced significantly by the presence of graphene nanoplates, since it may promote the PLA nucleation. This phenomenon was also reported by Wang and Qui (2011), who used graphene oxide and PLA [39]. Moreover, they observed that this behavior was related to the graphene oxide concentration in the polymeric matrix. PLA and PLA-nCD were used to produce electrospinning to exploit the numerous advantages of interconnected microfibers [14,40,41,42,43,44].

Within this framework, electrospinning has been defined by the European Commission as a key enabling technology since it offers a facile, low-cost, scalable, and highly controllable technique to produce fabrics with enhanced properties due to the fibers’ size. These mats can be applied for making filters, dressing for wounds, and smart fabrics thanks to their superior features. 

Moreover, the texture of the fabrics can impact the softness, the flexibility, the absorbency, and the conformability to different areas. Since a certain amount of chain entanglement is needed to keep the solution jet coherent during electrospinning, rheological analyses were performed to assess the concentration regimes of PLA and PLA-nCD solutions to set-up the manufacturing conditions of PLA and PLA-nCD fabrics and evaluate the fiber formation.

The dynamic rheological responses of neat PLA and PLA-nCD, dissolved in DCM at various concentrations, are reported. Samples showed a Newtonian plateau in this range of shear rate as also observed by McKee et al. 2004 for linear polyester polymer. A minimal slope of the shear thinning domain was observed in Figure 2; however, shear thinning may be revealed by extending the shear rate range [45].

A zero shear rate viscosity was obtained, and values corresponding to the composite polymer were higher than those of neat PLA. This was attributed to the presence of the high-modulus nCD that are homogenously dispersed and enabled to firmly repress the shear flow of the PLA moieties. The polymer contribution to the η_0_ was studied by defining the specific viscosity (η_sp_) and by plotting the logarithmic values of the specific viscosity and PLA-nCD concentration; indeed, the semidilute unentangled regime, semidilute entangled regime, and concentrated regime were identified. For neutral, linear polymers in a good solvent, η_sp_ ∼ C^1.0^ in the dilute regime, η_sp_ ∼ C^1.25^ in the semidilute unentangled regime, η_sp_ ∼ C^4.8^ in the semidilute entangled regime, and η_sp_ ∼ C^3.6^ in the concentrated regime. Herein, the dilute concentration regime, characterized by polymer chains dispersed at random, was not studied, since in this domain no inter-chain entanglements and indeed no electrospinning process occurred. According to Figure 3, the semidilute unentangled regime η_sp_ ∼ C^1.3^ agreed with the theoretical predictions (η_sp_ ∼ C^1.25^). In the semidilute entangled regime, the concentration exponent was 4.6, in good agreement with the theoretical scaling law exponent of 4.8. In the concentrated regime, the specific viscosity should scale with the PLA-nCD concentration to about the 3.6 power; however, weaker dependence was revealed (η_sp_ ∼ C^2.9^). It can be speculated that testing more PLA-nCD solutions with a concentration higher than 20% *w/v* may improve the goodness-of-fit.

From these results, a concentration of 20% *w/v* was selected to obtain beads-free fibers.

Finally, the effect of the binary solvent mixture (MC:DMF, 90:10 v:v) was evaluated, highlighting no differences in rheology with respect to MC alone. Although DMF possesses a high boiling point (~153 °C), it has a dielectric constant of 36.7 that permits the rise of the dielectric constant of the solution, and thus it can improve the fibers’ formation [46]. From the rheological point of view, no differences were pointed out between PLA and PLA-nCD solutions prepared by MC and MC:DMF. After that, the process parameters, namely, voltage and flow rate, were set-up and their effect on mean diameter, porosity percentage, and mean pore area were figured out. At first instance, it is known that voltage can affect the fiber morphology. Increasing the applied voltage and indeed increasing the electric field strength contribute to the enhancement of the electrostatic repulsive force on the fluid jet, which favors the thinner fiber formation. However, if the voltage extent exceeds the solution it is removed from the capillary tip more quickly, as the jet is ejected from the Taylor cone leading to a bigger fiber diameter [47]. From the results, the variation of the voltage between 20 and 25 kV did not affect the mats’ characteristics. Otherwise, the flow rate affected both fibers’ mean diameters and mean pore areas (Figure 5). Empirical observations indicate that the smallest diameters occur at the lowest flow rates, and these data are in agreement with Fridrikh et al. (2003), who predicted that the terminal diameter of the whipping jet was controlled by the flow rate, electric current, and the surface tension of the fluid; more in detail, they reported that the diameter changed with the scaling law exponent of 2 [48]. Therefore, the selected manufacturing parameters were 20 kV voltage and 0.3 mL/h. The effectiveness of the manufacturing set-up was confirmed by the production of textured prototypes composed by well-interconnected and uniform fibers with a smooth surface without imperfections, where nCD appeared to be well-included (Appendix A). The prototypes’ morphological features summarized in Table 3 play a vital role in proper moisture maintenance and gas exchanges between different compartments. As general consideration, nCD and DMF contributed to reduce the fiber thickness due to an increase in the electric conductivity of the polymer solution. As known from the literature, the quality of the obtained fibers is improved by increasing the conductivity that facilitates the Taylor cone formation [44].

The texture significantly impacted and improved the fibers’ alignment primarily for square and diamond patterns where more than 50% of the total fibers had the same orientation. In addition, the inclusion of nCD into the polymer fibers positively impacted the fibers’ alignment. This result can gain great interest, since few publications have reported on the relationship between the working parameters and the alignment effect of electrospun fibers; moreover, most of these works were performed through a mandrel rotating collector [46,49,50]. 

Finally, the precise measurement of the mechanical properties of the microfibrous matrix was reported. These tests gained insights into the tensile features of PLA and PLA-nCD upon heating. They are potentially useful considering the thermal conductivity and the reinforcement skills of nCD [26] and thermoplastic properties of PLA. Beyond that, the effect of the texture was investigated. At room temperature, nCD confirmed the marked improvement in elastic modulus and yield strength [51,52] for all the prototypes tested if compared to the neat polymer counterparts. Further, the flat mat showed the best mechanical performance compered to texturized mats. Heating the samples near but below the Tg of PLA for 3 h, the elastic modulus mainly increased in the composite mats, and the elongation performances were improved for texturized prototypes with respect to the flat mat (Figure 8). These results highlight that, upon heating, the reinforcing action of nCD is more efficient on the aligned fibrous mats. It is indicative that fabric undergoes elastic deformation more easily, returning to its original shape when the load is removed. However, the elastic modulus enhancement was accompanied by compromised ductility that allows a premature fiber fracture (Figure 7B); surprisingly, fabrics characterized by the highest percentage of aligned fibers (square and diamond) showed comparable breaking point values comparable to their relatively unannealed counterparts, indicating an unmodified resistance to tensile stress. The prolonged incubation up to 6 h determined an important fall of the elastic features, and similar behavior was observed at 69 °C (for both 3 h and 6 h of the incubation). This result may be attributed to the PLA chain relaxation and initial thermal polymer degradation since the PLA degradation sped-up the temperature [12]. Xu and coworkers [53] demonstrated this in aqueous buffers: they showed that PLA’s degradation rate was 4-folds faster in 37 °C in comparison to 25 °C, reaching the full degradation after 400 h, and at 60 °C the full degradation occurred more quicky.

Since PLA is the most used biodegradable polymer in clinical applications today, the potential biomedical application of the composite material was studied through a preliminary degradation test carried out by incubating the textured fabrics in DMED with serum (both PLA and PLA-nCD) up to 30 d. 

The samples revealed great fluid handling capacity over time, since fabrics enabled to at least duplicate their mass upon contact with a fluid. Likewise, our previous work for nCD enriched PLA–PCL electrospun fibers [26], and the hydrophobicity of nCDs decreased the water absorbency power of the composite mats. This behavior can be also correlated to the nCD role in the nucleation of polymer during crystallization. It was reported that nCD sharply increase the crystallinity of the polymer, and thereby polymer chains are more packed in the crystalline domains, and water penetration in the mats texture is slowed down [23].

Moreover, it was revealed that the matrices maintained their integrity over time; at the 30th day of incubation, a minimal mass loss (~10%) and slight PLA Mw reduction (15–20%) were recorded accordingly with the median half-life of PLA of 30 weeks [54]. The addition of nCD did not affect the biodegradability in the first 30 d. It should be mentioned that more pronounced effects of nCD on the polymer degradation may be observed at a longer incubation time, since the crystalline domains associated with the hydrophobic filler were more resistant to degradation than the amorphous region of the polymer [26,55].

Finally, the presence of toxic degradation products in the incubation media was excluded by testing them on NHDFs. 

These results are important for the biomedical and pharmaceutical application of polymeric matrices, of which an important property is the load-bearing capacity under external loading, and the implants should possess sufficient mechanical integrity until tissues repair, typically 5–12 weeks [56].

## 5. Conclusions

Even though PLA is more than 150 years old, it remains a fruitful platform for innovation in different fields such as biomedical, sensoring, and filtration areas. In this work, the electrospinning manufacturing of PLA-nCD composite mats was systematically set-up, and the produced fabrics were characterized. The set-up of the electrospinning process was mainly based on the identification of PLA-nCD concentration regimes. Entanglement concentration (Ce) and concentrated regime (C**) were graphically determined, and they were around 8% *w/v* and 20% *w*/*v*, respectively. As concerns the electrospinning parameters, a low feed rate (0.1 mL/h) downsized the fiber’s mean diameter and the fabric’s mean porosity area. The presence of nCD as smart fillers to improve the fabric’s mechanical, thermal, and electric features was combined with an innovative texture obtained by a patented process PCT/IB2020/05926. The texture and the addition of nCD improved the fibers’ alignment primarily for square and diamond patterns where more than 50% of the total fibers had the same orientation. A tensile analysis at RT revealed the reinforcement effect of nCD regardless of the fabric’s topology; instead, upon heating, the reinforcement was more evident for prototypes with aligned fibers.

The resulting prototype was flexible, highly absorbent, and conformable to the different body areas. In contact with a fluid, the fabrics revealed high fluid handling; the water absorption capability of microfibers with nCD (WU in a rank of 397%–637%) was lower with respect to the neat PLA (WU among 577–780%). Moreover, the addition of nCD into PLA fibers slowed down the fabrics’ degradation (ML ~10%). Toxic degradation products’ release was excluded since NHDFs treated with the degradation media showed a viability always higher than 70%. 

Further characterization can be useful and driven for a specific application.

## Figures and Tables

**Figure 1 polymers-14-05415-f001:**
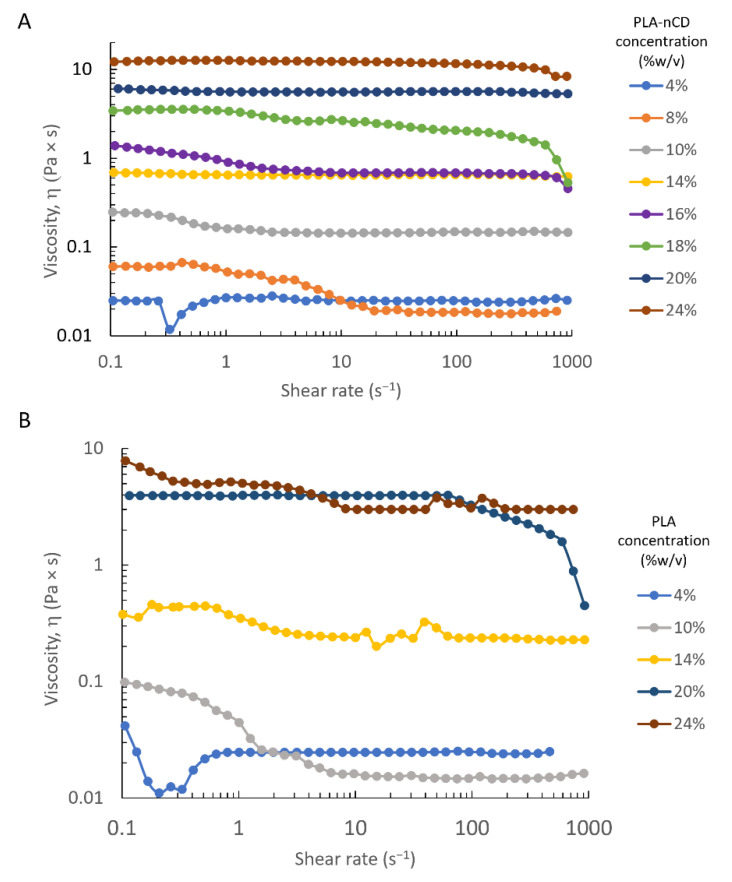
Dependence of viscosity on shear rate for PLA-nCD (**A**) and PLA (**B**) at various concentrations (4–24% *w*/*v*) by using DCM as solvent.

**Figure 2 polymers-14-05415-f002:**
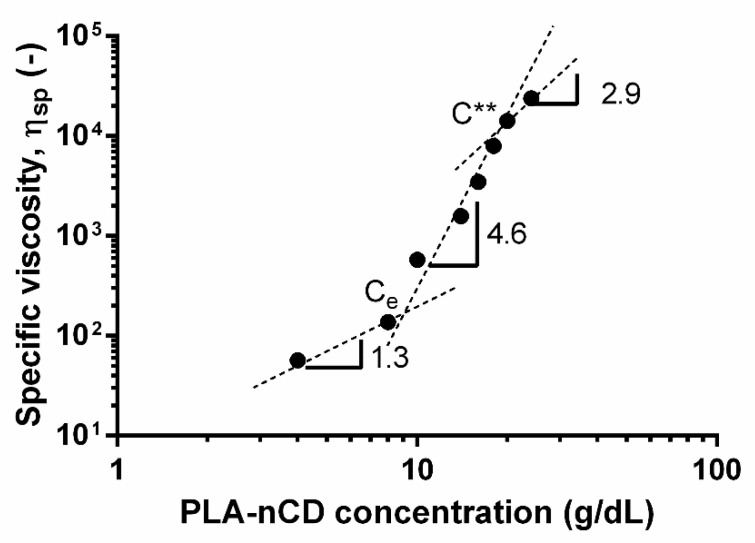
Dependence of specific viscosity on concentration for PLA-nCD. C_e_ indicates the entanglement concentration; C** is concentrated regime.

**Figure 3 polymers-14-05415-f003:**
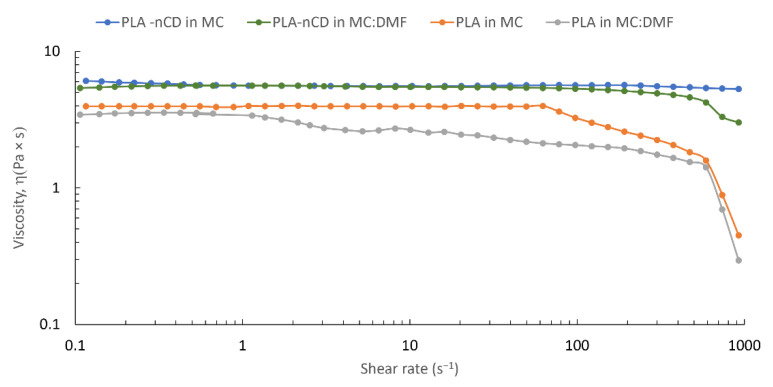
Effect of the solvent system (i.e., MC and MC:DMF) on the shear rate viscosity of PLA and PLA-nCD solutions (concentration = 20% *w*/*v*).

**Figure 4 polymers-14-05415-f004:**
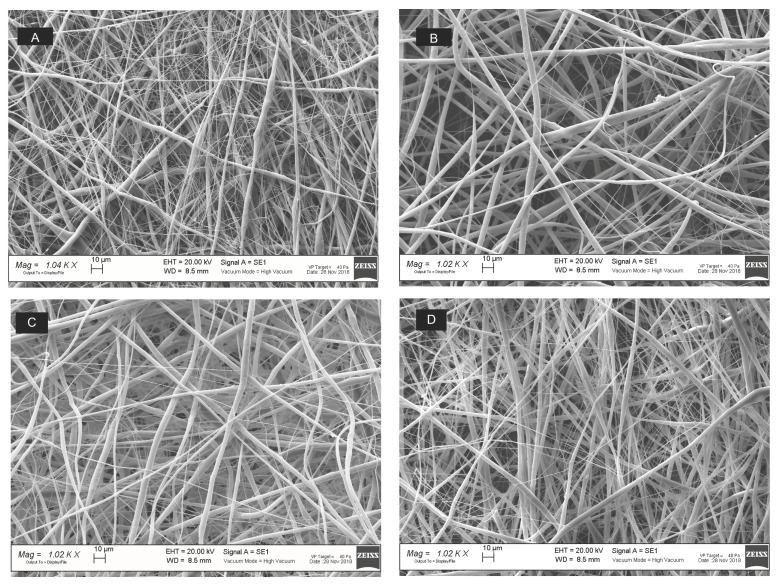
SEM micrographs (1 k× magnification, scale bar = 10μm) of PLA-nCD fabrics produced by setting the electrospinning parameters at 20 kV voltage and 0.1 mL/h feed rate (**A**), 25 kV voltage and 0.1 mL/h feed rate (**B**), 20 kV voltage and 0.3 mL/h feed rate (**C**), and 20 kV voltage and 0.3 mL/h feed rate (**D**).

**Figure 5 polymers-14-05415-f005:**
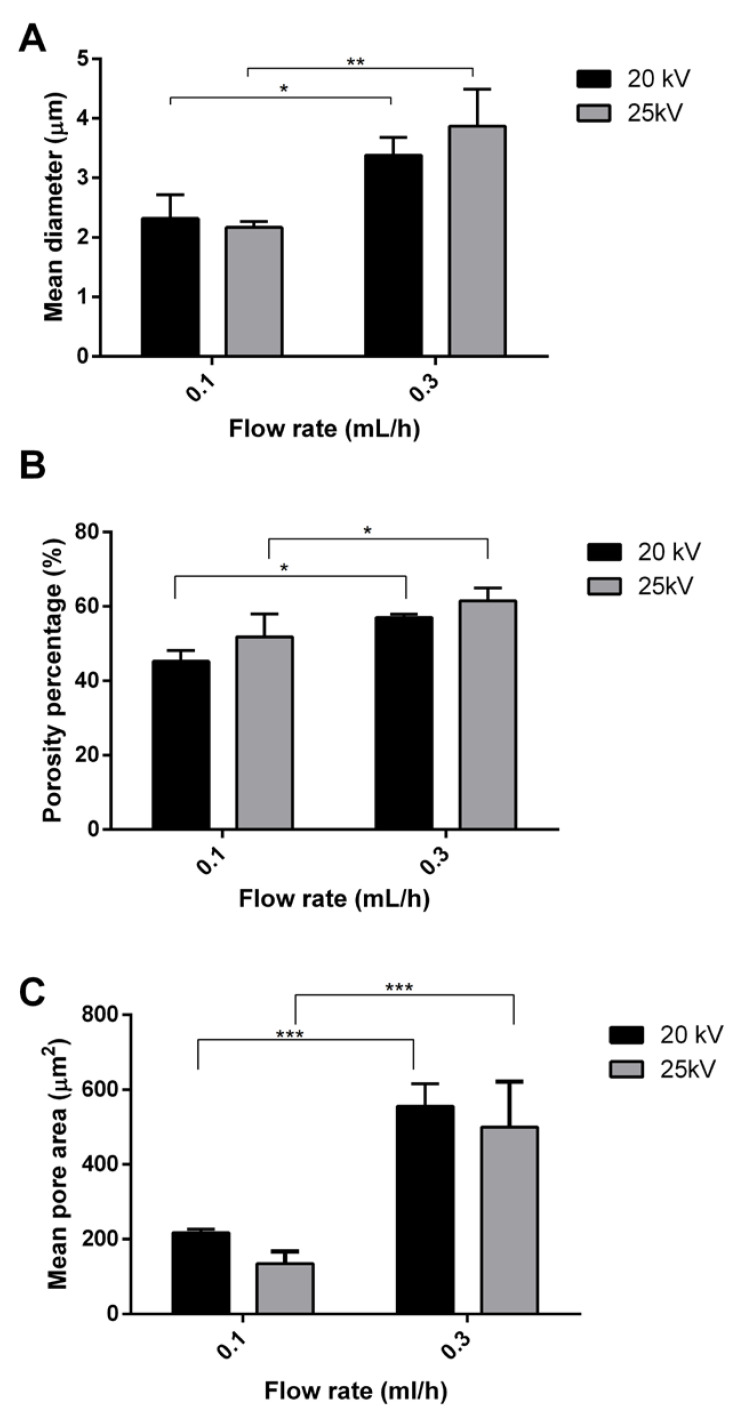
Effect of the electrospinning process parameters on fibers’ mean diameter (**A**), porosity percentage (**B**), and mean pore area (**C**). Multiple comparison test revealed significant differences for (*) *p* value < 0.05, (**) *p* value < 0.01 and (***) *p* value < 0.001.

**Figure 6 polymers-14-05415-f006:**
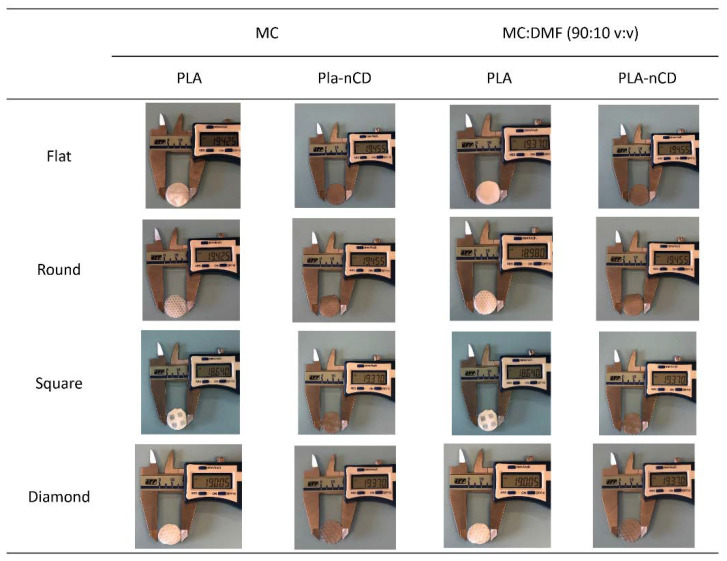
Macro details of PLA and PLA-nCD fabrics’ topology (flat, round, square, diamond) produced by two different solvent systems (MC, MC: DMF).

**Figure 7 polymers-14-05415-f007:**
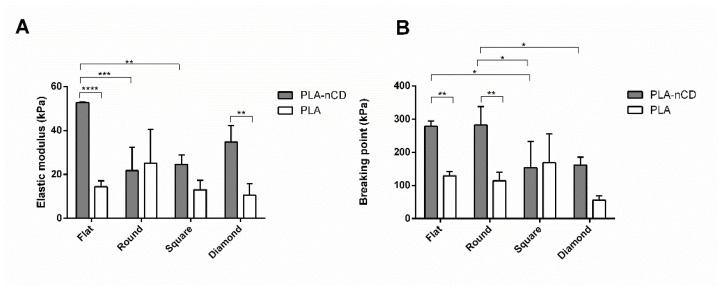
Elastic modulus (**A**) and breaking point (**B**) values of PLA-nCD (grey bar) and PLA fabrics (white bar) incubated at RT. Multiple comparison test revealed significant differences for (*) *p* value < 0.05, (**) *p* value < 0.01, (***) *p* value < 0.001 and (****) *p* value < 0.0001.

**Figure 8 polymers-14-05415-f008:**
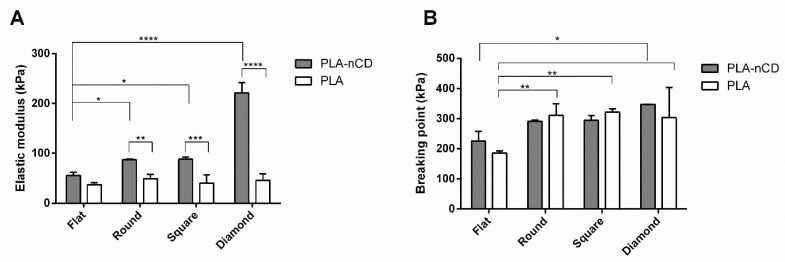
Elastic modulus (**A**)and breaking point (**B**) values PLA-nCD (grey bar) and PLA (white bar) fabrics incubated at 40 °C for 3 h. Multiple comparison test revealed significant differences for (*) *p* value < 0.05, (**) *p* value < 0.01, (***) *p* value < 0.001 and (****) *p* value < 0.0001.

**Figure 9 polymers-14-05415-f009:**
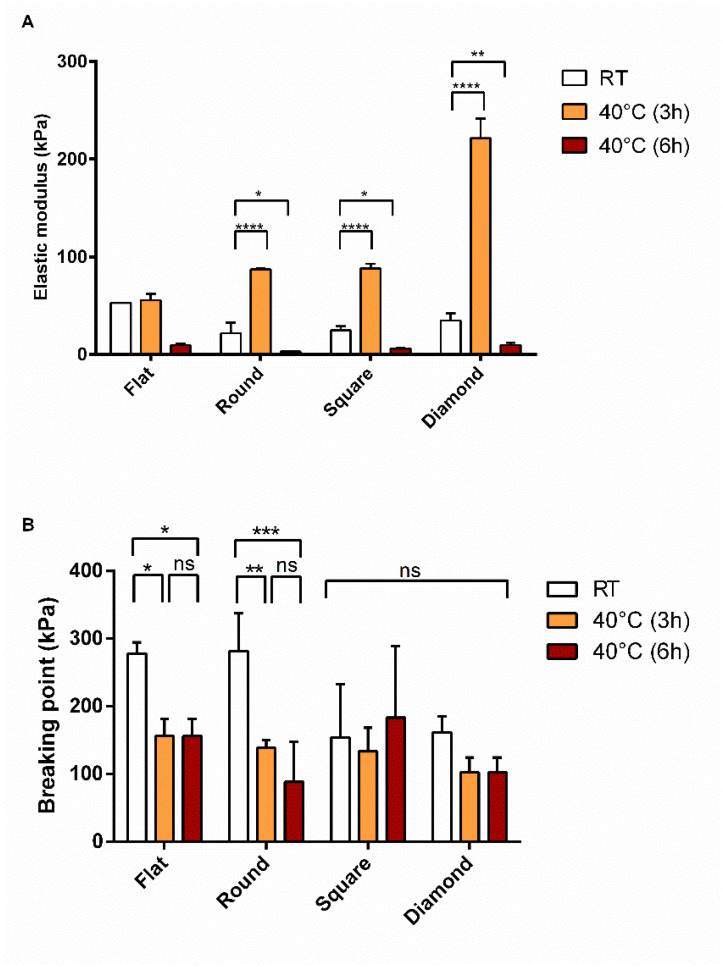
Elastic modulus (**A**) and breaking point (**B**) of PLA-nCD fabrics incubated at RT (white bar), 40 °C for 3 h (orange bars), and 6 h (red bars). Multiple comparison test revealed significant differences for (*) *p* value < 0.05, (**) *p* value < 0.01, (***) *p* value < 0.001 and (****) *p* value < 0.0001 while (ns) indicates no significant differences.

**Figure 10 polymers-14-05415-f010:**
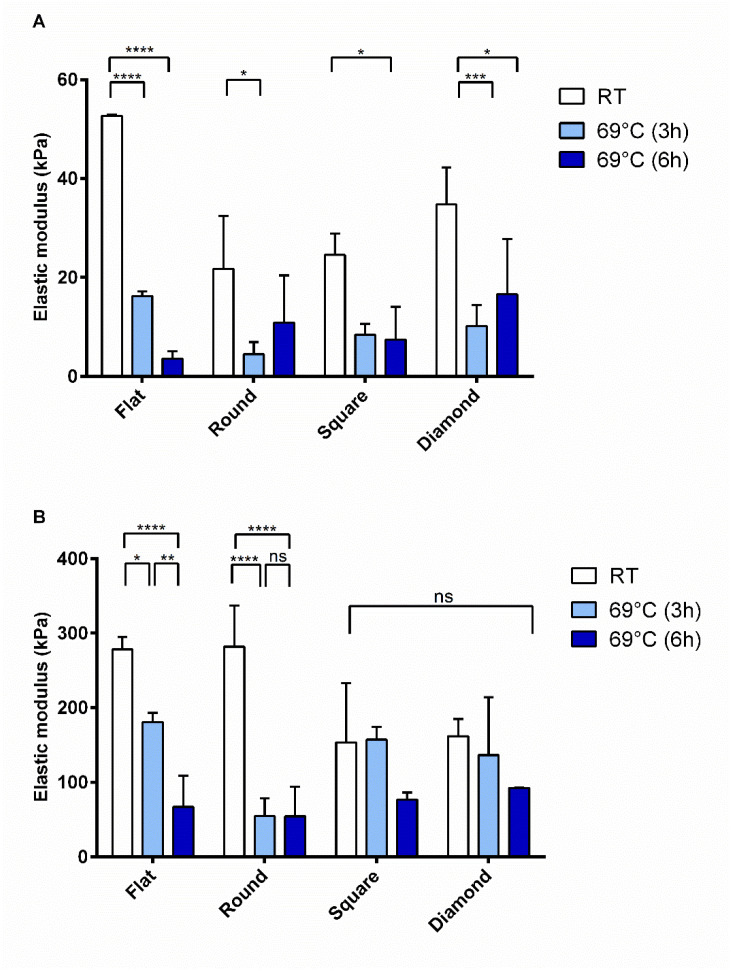
Elastic modulus (**A**) and breaking point (**B**) of PLA-nCD fabrics incubated at RT (white bar), 69 °C for 3 h (light blue bars), and 6 h (blue bars). Multiple comparison test revealed significant differences for (*) *p* value < 0.05, (**) *p* value < 0.01, (***) *p* value < 0.001 and (****) *p* value < 0.0001 while (ns) indicates no significant differences.

**Figure 11 polymers-14-05415-f011:**
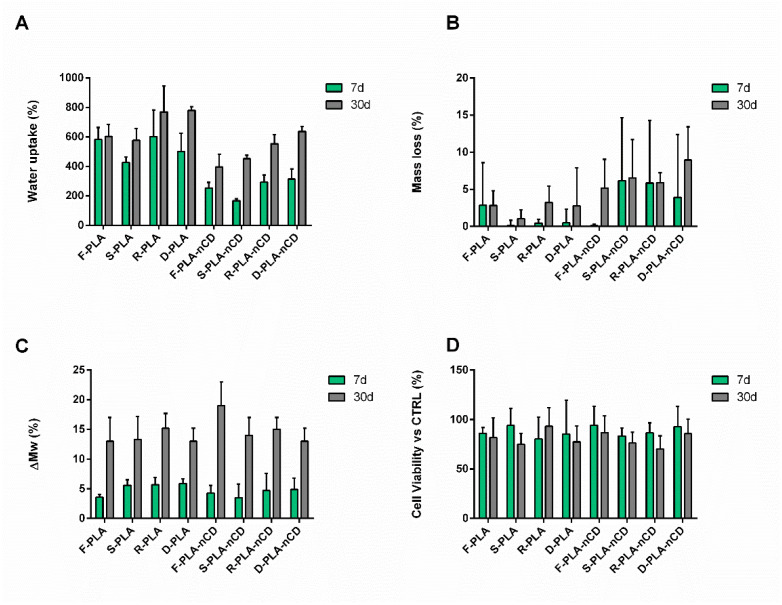
Water adsorption (**A**); mass loss (**B**); molecular weight variation, ΔMw% (**C**); cell viability after treatment with degradation media (**D**) of PLA and PLA-nCD fabrics incubated in DMEM up to 7d (green bar) and 30d (grey bar).

**Table 1 polymers-14-05415-t001:** Thermal data assessed by differential scanning calorimetry (DSC) measurements (second heating) for neat (PLA) and composite matrices (PLA-nCD).

Sample	T_g_ (°C)	T_cc_ (°C)	T_m1_ (°C)	T_m2_ (°C)	ΔH_cc_ (J/g)	ΔH_m1_ (J/g)	ΔH_m2_ (J/g)
PLA	56.7	104.8	146.8	155.3	28.6	11.9	18.0
PLA-nCD	56.9	98.0	144.8	153.6	22.7	9.3	18.8

**Table 2 polymers-14-05415-t002:** Zero shear rate viscosity values of PLA and PLA-nCD.

Concentration (%wt)	Zero Shear Rate Viscosity (Pa × s)	
	PLA	PLA-nCD
4	0.021	0.025
10	0.098	0.248
14	0.402	0.680
20	3.95	6.09
24	7.87	10.3

**Table 3 polymers-14-05415-t003:** Textured fabrics’ characterization: mean diameter, percentage of fibers with equal orientation, and porosity.

		Mean Diameter (µm)	Oriented Fibers (%)	Porosity (%)
Methylene chloride (MC)	F-PLA-1	3.35 ± 0.44	30	45 ± 3.0
	R-PLA-1	4.64 ± 1.22	33	30 ± 1.1
	S-PLA-1	3.85 ± 0.32	36	31 ± 2.6
	D-PLA-1	4.89 ± 0.84	27	21 ± 3.2
	F-PLA-nCD-1	2.85 ± 0.10	28	50 ± 4.1
	R-PLA-nCD-1	3.21 ± 0.41	23	52 ± 1.3
	S-PLA-nCD-1	2.88 ± 0.04	25	48 ± 1.4
	D-PLA-nCD-1	3.27 ± 0.05	53	58 ± 5.7
MC:DMF (90:10 v:v)	F-PLA-2	2.73 ± 0.02	26	50 ± 2.8
	R-PLA-2	3.13 ± 0.62	31	35 ± 3.3
	S-PLA-2	2.74 ± 0.16	71	50 ± 1.9
	D-PLA-2	2.60 ± 0.02	48	48 ± 2.1
	F-PLA-nCD-2	2.60 ± 0.16	37	43 ± 1.8
	R-PLA-nCD-2	2.66 ± 0.03	52	47 ± 2.7
	S-PLA-nCD-2	2.60 ± 0.25	63	46 ± 3.7
	D-PLA-nCD-2	2.68 ± 0.16	61	41 ± 5.0

## Data Availability

Data will be made available on request.

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
