# Peer review of "Graphene Nanoplatelets-Based Textured Polymeric Fibrous Fabrics for the Next-Generation Devices"

_polymers, 2022, doi:10.3390/polym14245415_

Round 1
Reviewer 1 Report
Several papers can be found in the literature dealing with electrospun fibers of PLA/graphene. For this reason the authors must refer in the introduction what new brings this manuscript. Also in the entire document the experimental results must be compared with the already published data.
Specific comments
The authors must refer the percentage %ww for nCD.
The plots of DSC measurements must be added at the text (first heating, cooling and second heating). The total melting enthalpy of PLA- nCD is larger than the crystallization enthalpy around 24%. How can be explained this?
According to figure 1, different concentrations are presented for PLA and PLA- nCD. Why? Also the y axes must be reduced in fig. 1, 3 (maximum value 20 or 10).
In fig. 2 the fitting in the concentration area 10-20% seems to refer to the area of 14-20%, although the area can be taken from 8-20%. Why?
The conclusion is too general without referring to the results and the discussion about them. What’s new findings from this work?
Author Response
Authors would like to thank the reviewers for the constructive suggestions and the opportunity to better implement our paper.
The Manuscript has been revised by the Authors according to the reviewers’ indications.
In the revised version of the Manuscript, changes have been highlighting using track changes.
A point-by-point response to the reviewers' comment suggestions has been generated by reporting Reviewers’ comments (in black) and Authors’ answers (in blue).
Reviewer #1
Several papers can be found in the literature dealing with electrospun fibers of PLA/graphene. For this reason the authors must refer in the introduction what new brings this manuscript. Also in the entire document the experimental results must be compared with the already published data.
Authors agree with the reviewer that there are several papers about electrospun fibers made of PLA and graphene; however, they showed controversial results. This topic has been underlined in the revised manuscript (line 71-91) and more references were added and discussed. Moreover, to the best of our knowledge this is the first attempt to combine the topology of PLA-nCD mats to the fabric’s properties considering a wide range of applications, the novelty of the work has been clearly defined in the revised manuscript (line 97-102).
Specific comments
The authors must refer the percentage %ww for nCD.
Authors used a commercially available filament (Grafylon®). Grafylon is the result of a collaboration between Filoalfa and Directa Plus companies and the nCD content declared by the manufacturer is 1% wt; as suggested by the reviewer, more details about raw material have been added in the revised manuscript.
The plots of DSC measurements must be added at the text (first heating, cooling and second heating). The total melting enthalpy of PLA- nCD is larger than the crystallization enthalpy around 24%. How can be explained this?
DSC measurements have been added. To not go overboard with figures, the DSC were added in Supplementary data.
The key difference between the melting and crystallization enthalpies is that melting enthalpy refers to the change in energy when a solid-state of a particular substance converts into the liquid state, whereas crystallization enthalpy refers to the heat that is either absorbed or evolved when one mole of a given substance undergoes crystallization.
The total melting enthalpy of PLA- nCD is larger than the crystallization enthalpy. Excluding the effect of the cooling rate that is equal for both samples, this phenomenon can be ascribed to the lower level of crystallization due to dynamic chain entanglements with nCD and not all of the molecular segments crystallise, defects are present; however, the presence of nCD facilitates the nucleation of crystals since the Tcc is reduced.
According to figure 1, different concentrations are presented for PLA and PLA- nCD. Why? Also the y axes must be reduced in fig. 1, 3 (maximum value 20 or 10).
As suggested by the reviewer the y-axes are reduced in Figures 1,3 of the revised manuscript. Same concentrations of PLA and PLA-nCD were tested namely 4% (blue line), 10% (grey line), 14% (yellow line),20% (dark blue line) and 24% (brown line) w/v to highlight possible differences in viscosity. For PLA-nCD, more solutions were tested to assess the different concentration regimes (Semidilute unentangled, Semidilute entangled, Concentrated regime).
In fig. 2 the fitting in the concentration area 10-20% seems to refer to the area of 14-20%, although the area can be taken from 8-20%. Why?
Authors thank for the clarification. Figure 2 has been revised indicating the correct fitting in the concentration area 8-20% and the main text has been properly modified. The entanglement concentration (Ce) has been found ˜8% w/v.
The conclusion is too general without referring to the results and the discussion about them. What’s new findings from this work?
The authors intent was to follow the Polymers guidelines for which section of conclusion is not mandatory but can be added to the manuscript if the discussion section is long or complex.
As suggested by the Reviewer, the most important results have been emphasized in the conclusion of the revised manuscript.

Reviewer 2 Report
The manuscript is about adding different concentrations of graphene nanoplatelets to the PLA and evaluating and comparing their physical properties with pure PLA. The solutions were used for electrospinning setup for making the electrospun web. The results confirmed that the presence of graphene nanoplatelets as smart fillers to PLA improved the fabric’s mechanical, thermal and electric features.
The Authors did valuable work. The manuscript is well-composed, written, and nicely discussed. In my opinion, the manuscript is suitable for publication in the Polymers journal after the completion of minor revisions. The comments that need to be addressed are given below:
1- In Figure 4 why do electrospinning conditions are changes, different conditions affect fiber diameters?
2- It is recommended to report the conductivity of mats with different concentrations of graphene nanoplatelets.
Author Response
Authors would like to thank the reviewers for the constructive suggestions and the opportunity to better implement our paper.
The Manuscript has been revised by the Authors according to the reviewers’ indications.
In the revised version of the Manuscript, changes have been highlighting using track changes.
A point-by-point response to the reviewers' comment suggestions has been generated by reporting Reviewers’ comments (in black) and Authors’ answers (in blue).
Reviewer #2
The manuscript is about adding different concentrations of graphene nanoplatelets to the PLA and evaluating and comparing their physical properties with pure PLA. The solutions were used for electrospinning setup for making the electrospun web. The results confirmed that the presence of graphene nanoplatelets as smart fillers to PLA improved the fabric’s mechanical, thermal, and electric features.
The Authors did valuable work. The manuscript is well-composed, written, and nicely discussed. In my opinion, the manuscript is suitable for publication in the Polymers journal after the completion of minor revisions. The comments that need to be addressed are given below:
- In Figure 4 why do electrospinning conditions are changes, different conditions affect fiber diameters?
At first instance, PLA-nCD flat fabrics (PLA-nCD-F) were electrospun starting from 20% w/v polymer solution and the influence of the process parameters as voltage and flow rate on the fabric’s morphology was evaluated.
Figure 4 reports one of SEM images taken for each combination of process parameters namely 20 kV voltage and 0.1 ml/h feed rate (4A), 25 kV voltage and 0.1 ml/h feed rate (4B), 20 kV voltage and 0.3 ml/h feed rate (4C) and 20 kV voltage and 0.3 ml/h feed rate (4D).
Images were elaborated to assess the mean diameter of the fibers, porosity percentage and mean pore area. Results were graphically depicted in Figure 6 and discussed in the text. From the results, it can be noted that low feed rate downsized the fiber’s mean diameter and the mean pore area of fabrics.
2- It is recommended to report the conductivity of mats with different concentrations of graphene nanoplatelets.
In this work thermal conductivity of the fabrics were not evaluated. In our previous work on nCD enriched PLA–PCL electrospun mats, fabrics showed good thermal properties. Excellent results were obtained for the electrospun PLA–PCL-based mats loaded with nCD (4% wt) that exhibited thermal conductivity of 1.27 ± 0.008 W/m K and thermal diffusivity of 1.07 ± 0.068 mm2 /s. It is worth noting that thermal conductivity value was significantly improving after the nCD addition compared to neat PLA PCL matrix (about 0.3 W/m K). This work (reference# 26) has been added and the thermal conductivity ability was mentioned (lines 499-501). Moreover, the electrical conductivity has been discussed since polymers, commonly insulators (10−10 S/m), became conductors when the nCD concentration reached a certain concentration into the composite material (conductivity increment of around 10 orders of magnitude).
References
Chiesa, E.; Dorati, R.; Pisani, S.; Bruni, G.; Rizzi, L.G.; Conti, B.; Modena, T.; Genta, I. Graphene nanoplatelets for the development of reinforced PLA-PCL electrospun fibers as the next-generation of biomedical mats. Polymers 2020, 12, doi:10.3390/polym12061390.
Stankovich, S.; Dikin, D.; Dommett, G.; Kohlhaas, K.; Zimney, E.; Stach, E.; Piner, R.; Nguyen, S.; Ruoff, R. Graphene-based composite materials. Nature 2006, 442, 282-286, doi:10.1038/nature04969.
Fang, C.; Zhang, J.; Chen, X.; Weng, G. Calculating the Electrical Conductivity of Graphene Nanoplatelet Polymer Composites by a Monte Carlo Method. Nanomaterials 2020, 10, doi:10.3390/nano10061129.

Round 2
Reviewer 1 Report
Author's reply can be accepted.